# Characterization of UVA-Induced Alterations to Transfer RNA Sequences

**DOI:** 10.3390/biom10111527

**Published:** 2020-11-08

**Authors:** Congliang Sun, Patrick A. Limbach, Balasubrahmanyam Addepalli

**Affiliations:** Rieveschl Laboratories for Mass Spectrometry, Department of Chemistry, University of Cincinnati, Cincinnati, OH 45221-0172, USA; sunca@mail.uc.edu (C.S.); limbacpa@ucmail.uc.edu (P.A.L.)

**Keywords:** UVR, photooxidation, tRNA, post-transcriptional nucleoside modifications, cusativin, RNA modification mapping, RNA oxidation

## Abstract

Ultraviolet radiation (UVR) adversely affects the integrity of DNA, RNA, and their nucleoside modifications. By employing liquid chromatography–tandem mass spectrometry (LC–MS/MS)-based RNA modification mapping approaches, we identified the transfer RNA (tRNA) regions most vulnerable to photooxidation. Photooxidative damage to the anticodon and variable loop regions was consistently observed in both modified and unmodified sequences of tRNA upon UVA (λ 370 nm) exposure. The extent of oxidative damage measured in terms of oxidized guanosine, however, was higher in unmodified RNA compared to its modified version, suggesting an auxiliary role for nucleoside modifications. The type of oxidation product formed in the anticodon stem–loop region varied with the modification type, status, and whether the tRNA was inside or outside the cell during exposure. Oligonucleotide-based characterization of tRNA following UVA exposure also revealed the presence of novel photoproducts and stable intermediates not observed by nucleoside analysis alone. This approach provides sequence-specific information revealing potential hotspots for UVA-induced damage in tRNAs.

## 1. Introduction

Transfer RNAs (tRNAs) deliver amino acids to the site of ribosome-mediated protein synthesis while decoding the messenger RNA (mRNA). This critical functional role is facilitated by the highly conserved folding of tRNA, which is essential for recognition and charging by aminoacyl-tRNA synthetases, interactions with the translational apparatus, and codon–anticodon pairing [1,2,3,4,5]. Post-transcriptional nucleoside modifications (PTMs) are ubiquitous in tRNA ranging from single atom substitution to complex hypermodifications catalyzed by elaborate enzymatic pathways [6]. They present varied nucleoside chemistries ranging from hydrophobic (aliphatic chains or aromatic substituents) to hydrophilic (including charged) functional groups in the anticodon and its loop [7,8]. These chemical groups facilitate translational efficiency, fidelity, and expansion of decoding approaches [7]. The chemical groups present elsewhere in the tRNA body (such as D-loop, TΨC-loop) provide functionalities for structural stabilization [9,10,11,12].

Levels of modified ribonucleosides can vary in response to stress exposure which can be due to alteration of levels of modifying enzyme [13,14] or tRNA abundance [15]. Such changes in modified tRNA are postulated to reprogram the codon-biased translation process [13,16], thus playing a critical role in regulating gene expression [15,17,18]. Furthermore, aberrant methylation of tRNA and ribosomal RNAs was reported under cellular stress, linking tRNAs [19] to stress response pathways, cell differentiation, and cancer [20]. A cell’s ability to cope with stress and stress responses could involve dynamic regulation of ribonucleoside modifications controlling the tRNA structure, function, metabolism, and mRNA translation [21].

We have an interest in understanding the degradative effects of ultraviolet radiation exposure (UVR) on tRNA and its nucleoside modifications. The solar UVR that reaches Earth’s surface contains 95% UVA (λ 315–400 nm) and 5% UVB (λ 280–315 nm) [22]. High incidence of UVA exhibits elevated photooxidative potential through reactions with cellular photosensitizers [23,24,25] via direct and indirect effects. Cellular exposure to UVA exerts photodynamic effects mediated by singlet molecular oxygen (^1^O_2_) and, to some extent, by hydroxy radical (OH**˙**) [26]. Endogenous photosensitizers such as cytochromes, flavin, heme, NAD(P)H, porphyrin, etc. get excited by UVA to cause photodamage [27] through two types of mechanisms [28,29].

Type I photosensitization involves charge transfer reactions between the triplet excited state of the photosensitizer and the target molecule in the vicinity to generate radical cations. Subsequent steps generate superoxide anion radical (O_2_^−^) and oxidation of guanine to 8-oxoguanine or Fapy-guanine [30,31] in nucleic acids. Type II photosensitization occurs through molecular oxygen leading to singlet oxygen which reacts with biomolecules containing double bonds (rich in electrons) including guanine, tryptophan, and histidine [32]. Oxidative lesions alter the base-pairing tendencies, leading to changes in genetic messages and functions [33,34,35]. Thiolated uracil, especially 4-thiouracil (4tUra) and 2,4-dithiouracil (2,4-dtUra), could serve as effective photosensitizers in both O_2_-rich and O_2_-depleted environments. Excitation of these nitrogenous bases in their pure form with near-visible radiation was found to generate a reactive triplet state and efficiently transfer its energy to molecular oxygen, generating 50% singlet oxygen [36,37]. They could also react directly with 5′-AMP or other nitrogenous bases in a mixture [36,38].

With the presence of RNA at levels 4-6 times higher than DNA [39], a higher level of RNA oxidation than DNA oxidation is expected [40]. In fact, causal links between RNA oxidation and neurodegenerative conditions such as Alzheimer’s disease (AD), Parkinson’s disease (PD), and amyotrophic lateral sclerosis (ALS) were observed in various studies [41,42,43]. Accurate detection and identification of the location of these oxidation products is necessary to establish the causality of such relationships. Oxidation reactions invariably lead to change in the mass of RNA building blocks, thus making them amenable for detection and quantification through mass spectrometry (MS).

Mass spectrometry, in combination with liquid chromatography, can accurately measure the changes in mass associated with the alterations of the chemical components such as addition of modifying groups to ribonucleosides [44,45]. The McCloskey group initially pioneered the identification and location of modified ribonucleosides in various RNAs [46,47,48] through a process referred to as RNA modification mapping [49]. In this approach, oligonucleotide digestion products resulting from a nucleobase-specific ribonuclease treatment are resolved on a liquid chromatography column and subjected to MS-based mass analysis following fragmentation through collision-induced dissociation (CID) of the oligonucleotide anions. Such CID fragment ions exhibit mass shifts corresponding to the mass values of modification groups, if present, and thus help reconstruct the modified RNA sequence. Because oxidation also entails a change in the mass of the ribonucleoside, our laboratory has utilized liquid chromatography coupled with mass spectrometry (LC–MS) to identify the susceptible chemical groups of ribonucleoside modifications and the resulting photoproducts upon UVA exposure [50]. Here, we document the hotspots of photooxidative damage and UVA-induced photoproduct formation in the sequence context through the application of RNA modification mapping procedures. Such information can help to understand the effects of photooxidative damage to tRNA on their decoding abilities and other cellular functions.

## 2. Materials and Methods

### 2.1. Materials

*Escherichia coli* tRNA and RNase T1 were procured from Roche Diagnostics (Indianapolis, IN, USA). *E. coli* tRNA^Tyr^ was obtained from Sigma-Aldrich (St. Louis, MO, USA). Riboflavin and all other chemicals were acquired from Fisher Scientific (Fairlawn, NJ, USA) unless otherwise specified. Ribonuclease cusativin was purified from the overexpression strain of *E. coli* on a nickel column using a His-tag protein purification kit from EMD Millipore (Burlington, MA, USA) as described [51].

### 2.2. Sample Preparation

An in vitro transcript (IVT) of *E. coli* tRNA^Tyr^ was prepared through T7-polymerase-mediated run-off transcription [52] of the linear tDNA^Tyr^ sequence using Ampliscribe from Epicentre (part of Lucigen, Middleton, WI, USA) and MegaClear kit from Ambion (Burlington, ONT, Canada) as per the manufacturers’ instructions. The tDNA^Tyr^ template was amplified by PCR using a combination of forward (5′-GGATCCTAATACGACTCACTATAGGGGTGGGGTTCCCGAGCGGCCAAA-3′) and reverse primer (5’-TGGTGGTGGGGGAA-3’) and high-fidelity *Pfu*Turbo DNA polymerase according to the manufacturer’s (Agilent Technologies, Santa Clara, CA, USA) guidelines. The forward primer contained the T7 promoter sequence (shown in italics) at the 5′-end to facilitate transcription. The transcript was confirmed by LC–MS analysis (data not shown).

### 2.3. UVR Exposure

The samples that were exposed to UVR included the in vitro transcript of tDNA^Tyr^, modified tRNA^Tyr^, and actively growing *E. coli* cells (see below). The in vitro transcript of tDNA^Tyr^ or modified tRNA^Tyr^ was irradiated with UVA (λ 370 nm peak wavelength, 360–400 nm range, 0.752 mJ cm^−2^ min^−1^) in the presence of 100 μM riboflavin (RF) under aqueous conditions for 20 min at room temperature (25 °C). Longer exposure time caused strand breakage in RNA. The UVA source was made in-house using NichiaSTS-DAI-2749D LEDs (Nichia, Wixom, MI, USA, catalog# 161031). Control (unexposed) tRNA samples were incubated in darkness in the presence of riboflavin for identical periods of time. The tRNA was precipitated with 0.3 volume of 7.5 M ammonium acetate and three volumes of ethanol overnight; the pellet obtained by centrifugation was subjected to 75% ethanol wash and dissolved in sterile water for subsequent digestion with ribonuclease T1 or cusativin.

### 2.4. E. coli Cell Culture and UVR Exposure

K-12 (obtained from the *E. coli* Genetic Stock Center) strain of *E. coli* was cultured in Luria Bertani medium to the mid-log phase (OD_600_ ~0.5). Two milliliters of resuspended cells were exposed to UVA for 1 h using an in-house built unit designed to hold 60 mm · 15 mm petri plates. The LEDs (LED Engin, Digi-Key Corporation, Thief River Falls, MN, USA, catalog # LZ1-00UV00) of this unit exhibited λ 370 nm peak wavelength, generating an energy of ~3 mJ cm^−2^ min^−1^. Exposure time longer than 1 h caused increased cell death.

### 2.5. Transfer RNA^Tyr^ Purification

The total RNA purified from exposed or unexposed *E. coli* cells by Tri-Reagent was used for isolation of small RNA (<200 nt) by salt fractionation as described before [53]. Transfer RNA^Tyr^ was purified from total tRNA using a biotinylated oligodeoxyribonucleotide (5′Biotin-d(TCCTTCGAAGTCGATGACGGCAGAT)-3′) that was complementary to the 3′-end of tRNA^Tyr^. The coupling of biotinylated oligodeoxynucleotides to streptavidin magnetic beads (New England Biolabs, Ipswich, MA, USA) and sequence-specific tRNA purification was carried out per the manufacturer’s protocol. Briefly, biotinylated oligonucleotides were incubated with binding buffer (0.5 M NaCl, 20 mM Tris-HCl, 1 mM EDTA) at 70 °C for 5 min and then chilled on ice. Subsequently, the DNA oligonucleotides were immobilized on magnetic beads at room temperature for 5 min, followed by incubation with total tRNA at 37 °C for 1 h with gentle shaking (~60 rpm). After washing the non-specifically bound RNA away, the bound tRNA^Tyr^ was released with elution buffer (10 mM Tris-HCl, 1 mM EDTA) at 70 °C for 5 min. The identity of isolated tRNA was confirmed by oligonucleotide sequencing through LC–MS.

### 2.6. Oxidized Oligoribonucleotide Characterization

The UVA-exposed and unexposed tRNA was enzymatically hydrolyzed to oligonucleotides with ribonuclease T1 or cusativin as described before [54]. Briefly, each µg of RNA (either untreated or treated) was mixed with ammonium acetate (110 mM, pH 6.0) and digested with either 25 units of RNase T1 or 1 µg of cusativin for 2 h at 37 ℃ or 60 ℃, respectively. Liquid chromatography–tandem mass spectrometry (LC–MS/MS) analyses were performed using a MicroAS autosampler, Surveyor MS Pump Plus HPLC system, and Thermo LTQ-XL (Thermo Scientific, Waltham, MA, USA) mass spectrometer equipped with an electrospray ionization (ESI) source. Reversed phase chromatography was performed on an Xbridge C18 1.0 × 150 mm (3.5 μm particle and 50 Å pore, Waters, Milford, MA, USA) column at 37 ℃ and 40 μL/min flow rate. Mobile phase A consisted of 200 mM HFIP, 8.15 mM TEA at pH 7.0 in water; mobile phase B consisted of 30% mobile phase A and 70% methanol. The gradient included the initial 5% mobile phase B for 2 min to enable sample loading, ramped to 20% at 10 min, 30% at 15 min, 70% at 62 min, and 95% at 67 min. After a hold of 5 min at 95% B, the column was equilibrated to the initial conditions (95% A, 5% B) for 18 min. In general, the sheath gas, auxiliary gas, and sweep gas at the ionization source were set to 25, 14, and 10 arbitrary units (au), respectively. The spray voltage was 4 kV, capillary temperature was 275 °C, and capillary voltage was −23 V, and the tube lens was set to −80 V. Data were recorded in negative polarity and profile mode using a zoom scan feature (scan range *m/z* 600 to *m/z* 2000) under scan event 1. This was followed by four data-dependent tandem mass spectral scans (scan events 2–5) triggered by the four most abundant oligonucleotide precursor anions from scan event 1. These molecular anions were subjected to collision-induced dissociation (CID) to yield sequence-informative fragment ions that had common 5′-end (c- or a-B ion series) or 3′-end (y- or w- ion series) in the tandem mass spectrometry (MS/MS) analysis to reveal the potential oxidation sites in the sequence.

### 2.7. Data Analysis

The template set of modified *E. coli* tRNA was obtained from the Modomics database [6]. The *m/z* values of the theoretically expected oligonucleotide digestion products with oxidations were computed using the Mongo Oligo Mass Calculator (https://mods.rna.albany.edu/masspec/Mongo-Oligo) and the custom-designed RNAModMapper [55]. The RAW data files were converted to MGF files using the MSConvert feature of Proteowizard for identification of the oligonucleotides through RNAModMapper. Its settings included mass tolerance of 1 Da for both precursor and fragment ions to include the isotope forms; the 3′ end was specified by 3′-PO_4_ for RNase T1 and/or 3′>PO_4_ (cyclic phosphate) for cusativin. The number of missed cleavages were set to 0 for T1 and 4 for cusativin. The thresholds of *P* score and dot product score were set at 70 and 0.8, respectively, as described before [56]. A fixed or variable sequencing approach was used with known modifications as reported for *E. coli* tRNA [6]. Scoring 80% of the expected CID fragment ions was required to denote the presence of oxidized or unaffected oligonucleotides. Manual processing of RAW data files was also performed for confirmation using the Qual browser feature of the Xcalibur software.

## 3. Results

### 3.1. Identification of the Susceptible Regions of tRNA to Photooxidation

To identify the regions of tRNA that are vulnerable to photooxidation, both modified and unmodified tRNA^Tyr^ (in vitro transcript) were exposed to UVA. These RNA molecules did not exhibit significant differences in their absorption spectra at wavelengths above λ 240 nm before or after UVA exposure. However, UVA exposure of RNA in the presence of riboflavin showed increased absorptivity at wavelengths below λ 240 nm (Appendix A). Except for thiolated nucleosides, no significant differences were observed on the levels of other modified nucleosides between unexposed and riboflavin-omitted but UVA-exposed samples in our previous studies [50]. Therefore, mapping of oxidation sites was restricted to UVA-treated RNA in the presence of riboflavin for in vitro exposure experiments. For the in vitro transcript, multiple oxidized guanosine-containing oligonucleotides were observed following digestion with ribonucleases T1 or cusativin and subsequent LC–MS/MS analysis. They contained a four-electron oxidation product of guanosine, 5-guanadinohydantoin (Gh, +6 Da), and were mapped to different regions of the sequence including the anticodon, variable, and TΨC (UUC) loops (Figure 1A). The observed oligonucleotides include U[Gh]UAAAUC>p *m/z* 1280.8 (*z* = −2) from the anticodon (position 33–40) (Appendix A), UGCC[Gh]UCAUC>p *m/z* 1054.2 (*z* = −3) (position 41–47)) (Appendix A) and AUC[Gh]AC>p at *m/z* 962.2 (*z* = −2, position 48–53) (Appendix A) from the variable loop, and UUCGAAG[Gh]UUC>p at *m/z* 1177.3 (*z* = −3, position 54–64) (Appendix A) in the UUC (TΨC) loop. The CID fragment ions in each of these cases exhibited corresponding mass shifts compared to unoxidized oligonucleotides, revealing the sites of oxidation. The levels of unoxidized versions of these oligonucleotides were drastically reduced in the UVA-exposed sample (data not shown). No such oxidized oligonucleotides were detectable in unexposed samples.

When modified tRNA^Tyr^ was exposed to UVA, oxidation products were also identified near the junction of acceptor and D-stems beside the anticodon and variable regions (Figure 1B). The modified tRNA^Tyr^ contains two 4-thiouridine [s^4^U] residues at position 8 and 9, and no signal was observed upon UV exposure for the oligonucleotide [s^4^U][s^4^U]CCCGp (*m/z* 960.4, *z* = −2) (Appendix A). This is accompanied with an increase in the signal for dethiolated oligonucleotide UUCCCGp (*m/z* 944.1), suggesting their photooxidative conversion. To confirm whether dethiolation happens under photooxidative conditions, we exposed an s^4^U-containing synthetic oligonucleotide, A[s^4^U]AG (*m/z* 1262.9, *z* = −1), to UVA before subjecting it to LC–MS analysis. We observed the formation of AUAG (*m/z* 1246.8), confirming the dethiolation by UVA-mediated photooxidation (Appendix A).

Manual processing of the LC–MS data revealed another oligonucleotide anion with a suggested sequence of FapyG[U]*[U]*CCCGp *m/z* 1124.3 (position 7–13) that had cross-linked uridines. The MS/MS of this molecular ion exhibited fragment ions that matched well with the expected mass values for formamidopyrimidine-Guanine (Fapy-G) ring-opened lesion at G7, dethiolation, and cross-linking (Figure 2A) of the oligonucleotide. Such a crosslinked product is expected to be formed by a mechanism illustrated in Figure 2B. The guanosine at position 7 adjacent to the two consecutive 4-thiouridine was converted to Fapy-rG, which is an advanced oxidation product of 8-oxo-guanosine [57]. Incidentally, the nucleoside analysis of UVA-exposed tRNA^Tyr^ revealed an abundant signal for *m/z* 302.110 (error = 3 ppm) corresponding to Fapy-rG, with the MS/MS confirming the presence of corresponding nitrogenous base (Appendix A). No signal for FapyG was observed in UVA-treated unmodified transcript (data not shown).

Only one nucleoside location, G44 in the variable loop, was found to be oxidized consistently between the in vitro transcript and modified tRNA^Tyr^ sequences. For the modified tRNA, this was revealed in the digestion product UGCC[Gh]UCAUCGAC>p (*m/z* 1380.8, (*z* = −3) position 41–52 (Appendix A). Another sequence location, position 34 of the anticodon loop, was also found to be susceptible in both tRNAs. During the tRNA processing inside the cell, G34 in tRNA^Tyr^ is modified to queuosine, Q. Similarly, the adenosine at position 37 is modified to 2-methylthio-*N*^6^-isopentenyladenosine [ms^2^i^6^A]. This oligonucleotide, U[Q]ψA[ms^2^i^6^A]AψC>p *m/z* 1397.4, corresponding to the anticodon region was drastically decreased (<10% abundance) (Appendix A). This is accompanied with an appearance of a new digestion product of a decreased mass (167.5 Da lower). The tandem mass spectrum of this molecular ion showed fragment ions corresponding to the sequence of U[*Q-99*]ψA[*ms^2^A*]AψC>p *m/z* 1313.6 (*z* = −2, position 33–40), indicating the photooxidative cleavage of queuosine and ms^2^i^6^A (Appendix A). Such products were also detected in the nucleoside analysis (Appendix A) [50] suggesting that they are not artifacts. Beyond these two locations, other Gh-containing digestion products (such as AUC[Gh]AC>p, UUCGAAG[Gh]UUC>p) observed for in vitro transcript tRNA^Tyr^ were not detected with the UVA-treated modified tRNA.

The observation of fewer Gh-containing oligonucleotides in the modified tRNA^Tyr^ compared to the unmodified version prompted us to perform comparative nucleoside analysis for guanosine oxidation products. The UVA-exposed unmodified transcript of tRNA^Tyr^ exhibited higher levels of Gh compared to its modified version (Figure 3A) suggesting that the presence of modified nucleosides could lower the overall photooxidative damage to the tRNA. However, the levels of 8-oxo-rG (two electron oxidation product) remained low compared to Gh (oxidation scheme shown in Figure 3B) for both unmodified and modified tRNA. These targeted ex vivo studies on tRNA^Tyr^ suggest that UVA damage is not randomly distributed throughout the tRNA sequence, but is rather focused to select regions that include the anticodon, variable loop, and s^4^U-containing regions at the junction of the acceptor and D stems.

### 3.2. UVA Exposure Effects on tRNA Under In Vivo (In Cellular) Conditions

While the above observed effects on tRNA nucleosides and their modifications represent the potential effects that could occur ex vivo, similar studies under cellular conditions (in vivo) are necessary to identify the most likely outcomes within growing cells. For this purpose, *E. coli* (K-12) cell culture at mid-log phase was exposed to UVA for 1 h, total tRNA was isolated, and tRNA^Tyr^ was purified with a complementary DNA probe. No riboflavin was added to the cell suspension. Cellular exposure for 1 h at 3 mJ cm^−2^ min^−1^ did not cause significant cell death (< 5%) (data not shown), although they exhibited delayed multiplication rates with prolonged lag phase when regrown with rich media (data not shown).

As before, the tRNA^Tyr^ isolated from UVA-exposed cells was analyzed using LC–MS/MS. The levels of the s^4^U-containing oligonucleotide, pGGUGGGG[s^4^U][s^4^U]CCC>p *m/z* 1338.3 (*z* = −3), were dramatically decreased (Appendix A). This decrease was accompanied with a concomitant increase of the dethiolated oligonucleotide, pGGUGGGGUUCCC>p *m/z* 1326.8 (*z* = −3), suggesting UVA-mediated dethiolation in these cells (Appendix A). A crosslinked uridine product along with FapyG that we observed under ex vivo reaction conditions was not detected in this sample. Within the anticodon, a hydrated form of queuosine (+18 Da) was observed in a cusativin digestion product with *m/z* 1406.4 (*z* = −2) exclusively in UVA-exposed cells. Tandem mass spectra of the molecular ion confirmed the sequence as U[Q+18]ψA[ms^2^i^6^A]AψC>p (Figure 4). However, this altered queuosine (Q + 18) was not detected in nucleoside analysis. Although the exact reason is not known, it is possible that this residue may not be stable under digestion or LC–MS analysis conditions of nucleosides or both. Furthermore, the oxidation product at G44 was also not observed from the variable region of purified tRNA^Tyr^. A list of UVA-induced alterations to tRNA^Tyr^ oligonucleotides is provided in Table 1.

### 3.3. Mapping UVA-Induced Effects on Other tRNAs

We have examined the status of other modified ribonucleosides that are susceptible to UVA-induced degradation [50] in the sequence context following exposure of *E. coli* cells. In these experiments, riboflavin was not added to the cell culture medium because the endogenous cytochromes, porphyrins, flavins, etc. act as photosensitizers inside the cell [27]. Initially, we probed the modification status of anticodon-containing oligonucleotides for other tRNAs for potential identification of photooxidative products formed upon UVA exposure. Comparative tRNA analysis of unexposed and exposed cells revealed interesting findings; certain modified oligonucleotides were found to be converted to their unmodified version following UVA exposure (Table 2). Due to challenges in identifying tRNA-specific changes in modification status, only those that arose within signature digestion products [58,59] are considered here.

A unique s^4^U-containing oligonucleotide, U[s^4^U]AACAAAG (*m/z* 1469.70, *z* = −2, position 7–15), found in tRNA^Cys^ exhibited decreased levels, while the level of unmodified version UUAACAAAG (*m/z* 1461.69, *z* = −2, position 7–15) increased following UVR exposure (Appendix A). The queuosine (Q) and t^6^A (*N*^6^-threonylcarbamoyladenosine)-containing oligonucleotide ACU[Q]UU[t^6^A]A[ψ]CCGp (*m/z* 1367.5, *z* = −3) from tRNA^Asn^ decreased and is accompanied by an increase in levels of signal that corresponded to queuosine+18 Da (*m/z* 1373.5, *z* = −3) in the UVA-exposed sample (Appendix A), which is consistent with the previous observations with tRNA^Tyr^.

The cmo^5^U (uridine 5-oxyacetic acid)-containing oligonucleotide CUU[cmo^5^U]G (*m/z* 829.1, *z* = −2, position 31–35), which is unique to *E. coli* tRNA^Ala^ (Figure 5), was decreased upon UVR exposure while the levels of oligonucleotide CUUUG (*m/z* 792.08, *z* = −2) increased. Similarly, mnm^5^s^2^U (5-methylaminomethyl-2-thiouridine) and m^2^A (2-methyladenosine)-containing oligonucleotide, CCCU[mnm^5^s^2^U]UC[m^2^A]CGp (*m/z* 1068.47, *z* = −3, position 30–39) from tRNA^Glu^ was observed at decreased levels in UVA-exposed cells with the concomitant increase of CCCUUUCACGp (*m/z* 1044.12, charge = −3, position 30–39) (Appendix A).

The modifications, cmo^5^U and mnm^5^s^2^U, were previously shown to be degraded oxidatively [50]. However, the loss of m^2^A was not evaluated in the previous work, and the data obtained here prompted us to reanalyze the nucleoside data. Analysis of the associated LC–MS data files revealed a mean decrease of 24% ± 10% with *p* value of 0.051 in the levels of m^2^A following UVA exposure of cells. We consider the observed *p* value at the borderline of statistical significance based on two-tailed Student’s *t*-test. One possibility is that m^2^A, which is present in various tRNAs such as tRNA^Arg^, tRNA^Asp^, tRNA^Gln^, tRNA^Glu^, and tRNA^His^, may not be equally accessible to photooxidative degradation. Examination of the signal abundance for m^2^A-containing oligonucleotides revealed a general decrease in levels for tRNA^Arg(ICG)^, tRNA^Gln^, and tRNA^His^ (Appendix A). The m^2^A-containing digestion product for tRNA^Asp^ was not detected in the analysis. The levels of oligonucleotide, [m^5^U]ψCGp, whose modifications are not known to be affected by UVA exposure in the previous studies [50], largely remained unaltered in UVA-exposed samples (5.7E7 vs. 5.9E7 in control) (Appendix A). Several oligonucleotides that contained UVA-susceptible modifications (such as k^2^C (2-lysidine) and cmnm^5^s^2^U (5-carboxymethylaminomethyl-2-thiouridine), t^6^A) exhibited decreased levels, but their potential conversion products were not identifiable in this analysis (data not shown).

## 4. Discussion

### 4.1. Hotspots of Photooxidative Damage in tRNA

LC–MS detects the changes in the mass values of ribonucleosides or oligonucleotides corresponding to the alterations of building blocks at the atomic level. Our previous studies on the effect of UVA exposure on tRNA revealed insignificant levels of oxidation products when tRNA was irradiated in the absence of riboflavin. However, the thiolated nucleoside levels were decreased by UVA under these conditions [50]. Since our main goal is to identify the effects of UVA on other nucleoside modifications, mapping of oxidation products was limited to the RNA that was exposed to UVA in the presence of riboflavin (in vitro exposure). On the other hand, cells do contain endogenous photosensitizers [27]; therefore, riboflavin was not added to the medium when cells were exposed to UVA. RNA exposed to UVA in the presence of riboflavin showed increased absorptivity at wavelengths below λ 240 nm (Appendix A); this characteristic behavior is similar to the observations with the UVA treatment of thiolated uracils [36]. Although some of the previously observed photooxidation of ribonucleoside modifications (that contain amino, -oxy, or sulfur groups) [50] could be directly attributed to specific positions of tRNA; the prevalence of guanosine oxidation in various regions of RNA cannot be revealed by nucleoside analysis alone. We opined that the RNA modification mapping approaches could be utilized to locate the RNA oxidation sites within the tRNA sequence.

The in vitro transcript of tDNA^Tyr^ does not contain any modified nucleosides; therefore, it may or may not assume the native structure of tRNA^Tyr^. Photooxidation can be expected to be random due to potentially equal accessibility if the sequence does not assume higher order structure. However, observation of higher levels of guanosine oxidation in the anticodon and variable regions (Figure 1A) and lack of detection of oxidative damage in other regions argues that the photooxidation sites are not random. This may imply that the unmodified transcript is assuming a structure that could restrict the photooxidative damage to anticodon and variable loop regions. Subjecting the modified tRNA^Tyr^ to identical treatment also resulted in expected dethiolation at the junction of amino acid acceptor and D-stems (Figure 1B). The lack of detectable levels of photooxidation in other regions in both unmodified and modified forms of tRNA^Tyr^ suggests that the anticodon, variable loop, and thiolated regions are highly prone to photooxidation. These regions could potentially serve as hotspots of UVA-induced damage when tRNA^Tyr^ is exposed to UVA outside the cell (Figure 1C).

The predominance of the 4-electron oxidation product of guanosine, Gh, over 8-oxoguanosine (8-oxorG) (Appendix A) is consistent with previous studies where inclusion of a photosensitizer increased levels of higher oxidation products of guanosine [50]. Although 8-oxodG is the major product formed by UV radiation in DNA, more than a dozen other lesions were observed as minor products depending on the pH, stacking interactions in the duplex, availability of other oxidants, nucleophiles, etc., presumably through various oxidation pathways. Of these, Gh lesions were observed at higher levels in duplex DNA under neutral pH [60]. It is possible that the initial oxidation product of guanosine, 8-oxoG or 8-oxodG, acts as a sink for electron hole trapping, leading to a second oxidizing event, thus accumulating more damage at the same site. This is because oxopurines such as 8-oxodG have lower redox potentials than the canonical base (1.29 V for dG vs. 0.74 V for 8-oxodG) [61], making it more sensitive to further oxidation. Since tRNA is a highly folded molecule with duplex regions in the secondary or tertiary L-shaped structure, there is a possibility that some regions of the sequence are more accessible to photooxidative damage than others. Observation of Gh in restricted sets of oligonucleotides corresponding to the anticodon and variable regions in both unmodified and modified tRNA^Tyr^ strongly support this view, indicating their vulnerability (i.e., hotspots) for photooxidative damage as depicted in Figure 1C. Dose-dependent UVA exposure conditions could confirm the possibility of 8-oxo-rG serving as a sink for electron hole trapping for the formation of Gh.

### 4.2. Modifications in tRNA can Influence the Pattern of Photooxidative Damage

UVA exposure (λ 365 nm, 45 kJ/m^2^) of monkey kidney cells (CV-1) that were grown in s^4^U media caused RNA–RNA crosslinks and RNA–protein bridges [62]. Moreover, s^4^U is widely used as a photoactivatable probe for RNA–protein and RNA–RNA for tertiary interaction studies [63]. Thiolated uracil, especially 4-thiouracil (along with 2,4-dithiouracil), is known to be photoreactive, and exposure of pure 4-thiouracil solution (24 µM) resulted in excitation, generation of ^1^O_2_, and reaction with 5-AMP (5′-adenosine monophosphate) in the mixtures [36,37]. Thus, the presence of crosslinks at the s^4^U positions in the current ex vivo exposure conditions are not entirely unexpected. Further, oxidation of neighboring guanosine may be attributed to the generation of reactive oxygen following excitation and triplet state conversion of s^4^U. Although the 4-5-crosslinked diuridine photoproduct observed in oligonucleotide analysis (Figure 2) was not detected in nucleoside data, its existence cannot be ruled out, as it is possible that this structure may not remain stable under RNA hydrolysis conditions or may not be retained on a chromatographic column as seen in other cases [64]. Moreover, such a crosslink is similar to the one observed between s^4^U (position 8) and cytidine (position 12 or 13) in bacterial tRNAs following UV absorption [65,66].

Dethiolation by chemical oxidant under ex vivo conditions was also reported by Nawrot’s laboratory research team. Their studies indicated guanosine-oxidation-independent dethiolation reactions of 2-thiouridine [s^2^U] to form 4-pyrimidinone nucleoside (H_2_U) and uridine when treated with 100 mM H_2_O_2_ [67,68], where the ratio of these oxidation products formed depended on pH and substitutions at C5 position of s^2^U [69]. They also noticed that cytochrome catalyzed H_2_O_2_-assisted dethiolation through uridine sulfenic (U-SOH), sulfinic (U-SO_2_H), and sulfonic (U-SO_3_H) intermediates [70]. The dethiolation accompanied with the neighboring purine oxidation (at least in some molecules) (Figure 2) in the current studies suggests interplay between the UVA, s^4^U, and the susceptibility of guanosine to oxidation.

Thiolated uridine exhibits an absorption spectrum extending up to near-UV (λ 300–400 nm range) [36,71,72] and is postulated to play protective function. This is because cells deficient in enzymes that make s^4^U are more easily killed by near-UV light in bacteria [71,72]. Such a possibility could also explain the observation of higher levels of Gh for unmodified tRNA compared to its modified version (Figure 3), where modified nucleosides could serve as easy targets for the reactive oxygen species (ROS) generated by the UVA exposure.

Increased photooxidation of queuosine and ms^2^i^6^A, at least in ex vivo conditions, raise the possibility of becoming easy targets for the UVA-induced ROS; modified nucleosides are also known to favor compact folding of the tRNA molecule, leading to reduced accessibility for ROS compared to unmodified tRNA. However, the failure to detect oxidized oligonucleotides in regions other than anticodon and variable loops in unmodified tRNA argues against the latter unless the in vitro transcript also assumes tertiary structure involving interactions between the V-loop and D-stem loop. Either way, the presence of certain post-transcriptional nucleoside modifications (PTMs) seemed to reduce the levels of guanosine oxidation either directly or indirectly in modified RNA (Figure 3).

### 4.3. Detection of the Photooxidative Products of Post-Transcriptional Nucleoside Modifications

The observation of two different types of photooxidative conversion products for queuosine following UVA exposure under in vitro (Q-99) (Appendix A) and in vivo (Q+18) (Figure 4) conditions might suggest different stages of photooxidative degradation. Observation of the Q-99 product in the nucleoside analysis supports the formation of such a product and predicts the potential cleavage point (based on mass value) on the modification group (Appendix A) by the UVA-induced ROS. The Q+18 product observed in oligonucleotide analysis could not be corroborated by nucleoside analysis. However, a similar product was discovered for the Q-containing oligonucleotide in tRNA^Asn^ (Appendix A), indicating that this may not be an artifact but a potential photooxidation product that could occur in Q-containing tRNAs. Although the final location of the water molecule (potentially as -OH and -H) cannot be established by this study, the double bond on the 5-membered ring of the queuosine side chain could be one possible destination. Further studies could evaluate the occurrence of such possibility.

Although the photooxidative products that exhibit altered mass shifts could be identified (Appendix A) by nucleoside analysis, identification of reaction products that result in loss of an entire modification is not possible by such analysis. This is because the high abundance of canonical nucleosides (rA, rG, rC, and rU) preclude the possibility of detecting subtle changes in their levels due to detector saturation. Such situations would require oligonucleotide analysis; this is demonstrated with a number of PTMs, including cmo^5^U (Figure 5), s^4^U (Appendix A), mnm^5^s^2^U (Appendix A), or m^2^A (Appendix A) in the current study. Although these changes can be attributed to the direct impact of UVA based on in vitro experiments including the synthetic oligomer (Appendix A), the possibility of the synthesis of new tRNA transcripts that stay unmodified during UVA exposure cannot be entirely ruled out under cellular conditions. Such a possibility of hypomodified RNA synthesis can be verified by dynamic nucleic acid isotope labeling coupled mass spectrometry (NAIL–MS) technique [73].

Out of the various modified nucleosides whose levels were adversely affected by UVA exposure [50], the photooxidation products of t^6^A, m^6^t^6^A (*N*^6^-methyl-*N*^6^-threonylcarbamoyladenosine), k^2^C, and ac^4^C (*N*4-acetylcytidine) were not identified in this analysis. However, as is the case of tRNA^Tyr^ in the current study, analysis of the purified isoacceptor tRNA (that contain these modification/s) could help identify the photooxidation products and intermediates induced by the UVA exposure. Alternatively, identification of partially degraded products in the nucleoside analysis could help identify the status in the sequence context through RNA modification mapping procedures. Nevertheless, the current studies have identified UVA-induced photooxidative products for Q, ms^2^i^6^A, cmo^5^U, mnm^5^s^2^U, and s^4^U both under in vitro and in vivo conditions in the sequence context.

### 4.4. Potential Consequences on tRNA due to Photooxidative Degradation of PTMs

Although the current studies do not address the exact impact of photooxidation on overall structure and function on tRNA, it is easy to predict an adverse effect on stability and function. The s^4^U at position 8 is thought to promote the stable tertiary structure of tRNA by reinforcing the base interactions in the D- and T-loops and inhibiting unfavorable conformation [74]. Absence of s^4^U decreased the melting temperature of tRNA^Ser^ [75] and promoted degradation through the RNA degradosome in bacteria [74]. Similarly, it may be speculated that guanosine oxidation at various locations could affect base-pairing [33,34,35] interactions, leading to adverse effects on conformational stability and function of the tRNAs in UVA-exposed cells.

Modifications at positions 34 and 37 in the anticodon loop coordinate the decoding efficiency and translation accuracy properties of tRNA by improving tRNA binding to codons during the translation process [76]. Loss of these modifications would be expected to adversely affect those functions during cellular translation, thereby making such tRNA targets for degradation. Indeed, the growth delay effect observed soon after UVR exposure could be related to the clearing of damaged tRNA [65] following recognition and binding by proteins such as polynucleotide phosphorylase in *E. coli* [77,78] and potentially YB-1 protein in humans [79] for subsequent turnover by RNA degradation machinery in *E. coli* [80,81] and the eukaryotic exosome [82]. However, further studies are required to understand the fate of cellular tRNA that has undergone partial degradation of PTMs and their relationship to the oxidation of canonical bases such as guanosine.

## 5. Conclusions

In the present study, we identified the susceptible regions of tRNA for UVA-induced photooxidative stress by locating the RNA oxidation sites through RNA modification mapping procedures. The oligonucleotide analysis indicated accumulation of photooxidative damage in anticodon and variable loop or regions containing UVA-sensitive modifications. These efforts identified the conversion products, and their locations were not revealed by nucleoside analysis alone. These studies also identified the modification groups that were completely degraded, where the status flipped from modified to unmodified upon UVA exposure. Our studies also raised interesting questions on the role of ribonucleoside modifications in reducing the oxidation levels induced by UVR exposure. Identification of the types of UVA-induced photoproducts and intermediates in tRNA in the current study can facilitate the understanding of their potential impact on cellular translation and gene expression.

## Figures and Tables

**Figure 1 biomolecules-10-01527-f001:**
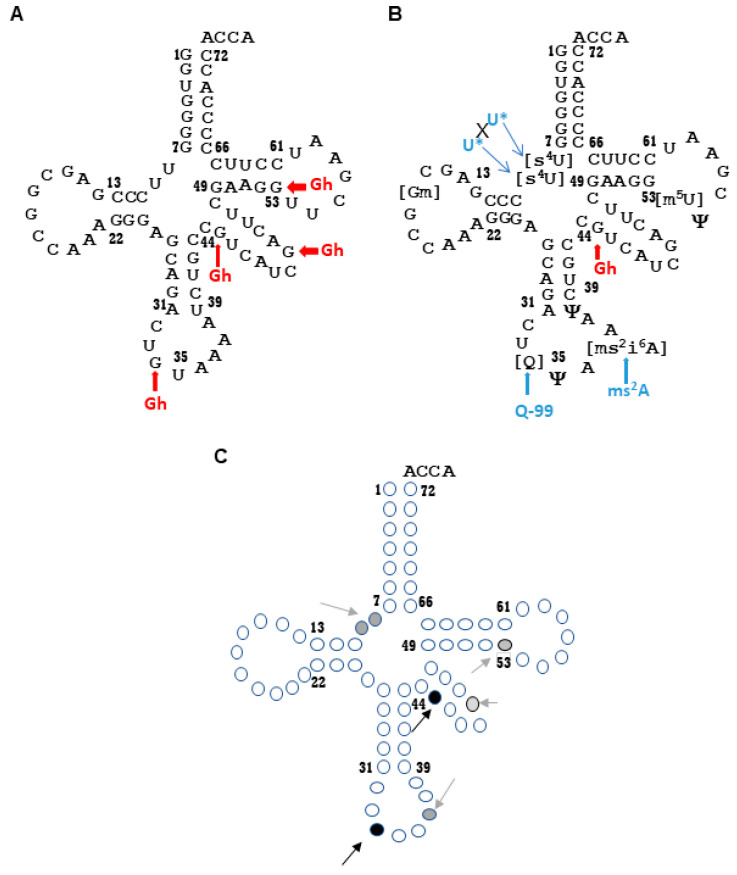
Hotspots of photooxidative damage in tRNA^Tyr^. Following UVA exposure in the presence of a photosensitizer, either (**A**) unmodified (in vitro transcript of tDNA^Tyr^) or (**B**) modified tRNA^Tyr^ was digested with RNase cusativin or T1 and subjected to liquid chromatography–tandem mass spectrometry (LC–MS/MS) analysis. Mapping the observed digestion products to the original sequence identified the locations of photooxidative damage accumulation. (**C**) Hotspots of UVA-induced photooxidative damage on a generic tRNA. Nucleotides are represented in circles. The most susceptible nucleotides in specific regions are shown as dark black circles; nucleotides that are susceptible due to specific modifications or variable susceptibility at specific locations are shown as grey circles. Gh: 5-guanadinohydantoin; U*XU*: crosslinked uridine; Q-99: queuosine-99 Da; ms^2^A: 2-methylthioadenosine.

**Figure 2 biomolecules-10-01527-f002:**
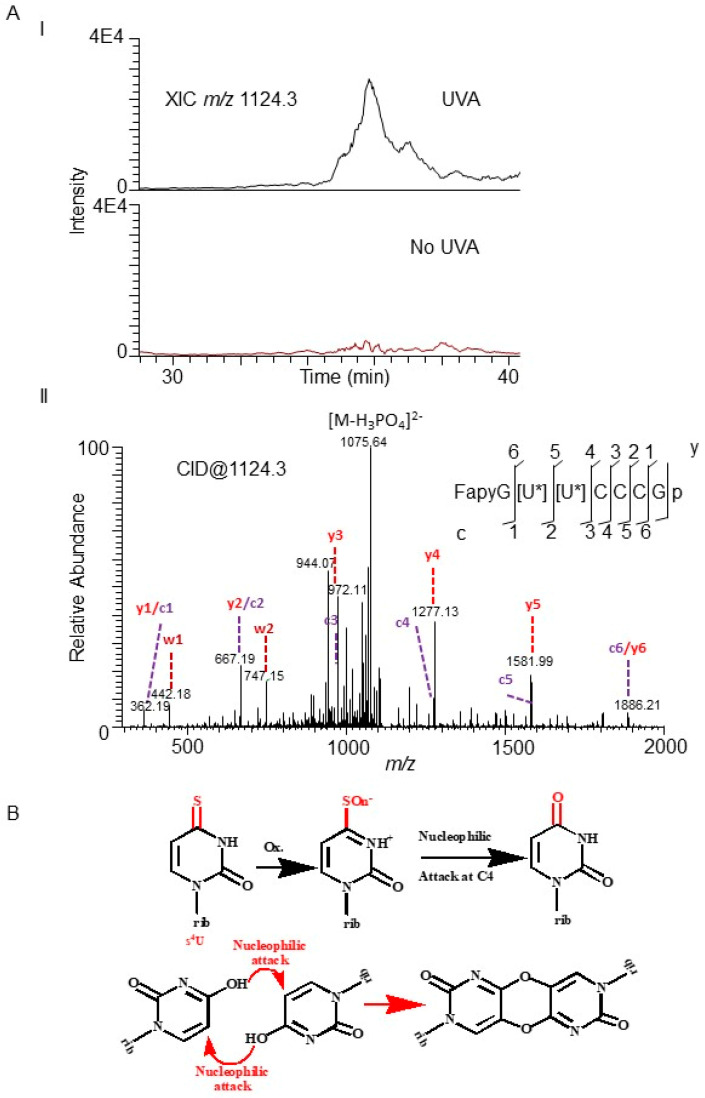
Detection of oligonucleotide containing diuridine crosslink and oxidized guanosine following UVA exposure. (**A**) LC–MS detection of the oligonucleotide FapyG[U]*[U]*CCCGp (*m/z* 1124.3, position 7–13) from UVA-exposed *E. coli* tRNA^Tyr.^ (I) Extracted ion chromatogram (XIC) for *m/z* 1124.3 in the exposed and unexposed samples. (II) Tandem (MS/MS) mass spectrum of *m/z* 1124.3 depicting the sequence informative product ion pattern (cn-ion series for 5′ and yn/wn ion series for 3′-end are depicted in purple and red color) of precursor oligonucleotide anion. [U*][U*]-diuridine crosslink, FapyG- formamidopyrimidine-Guanine (**B**) Potential mechanism of formation of 4–5 cross-linked diuridine photoproduct. The chemical group -SO_n_^−^ represent either sulfenic (U-SOH) or sulfinic (U-SO_2_H) or sulfonic (U-SO_3_H) intermediates.

**Figure 3 biomolecules-10-01527-f003:**
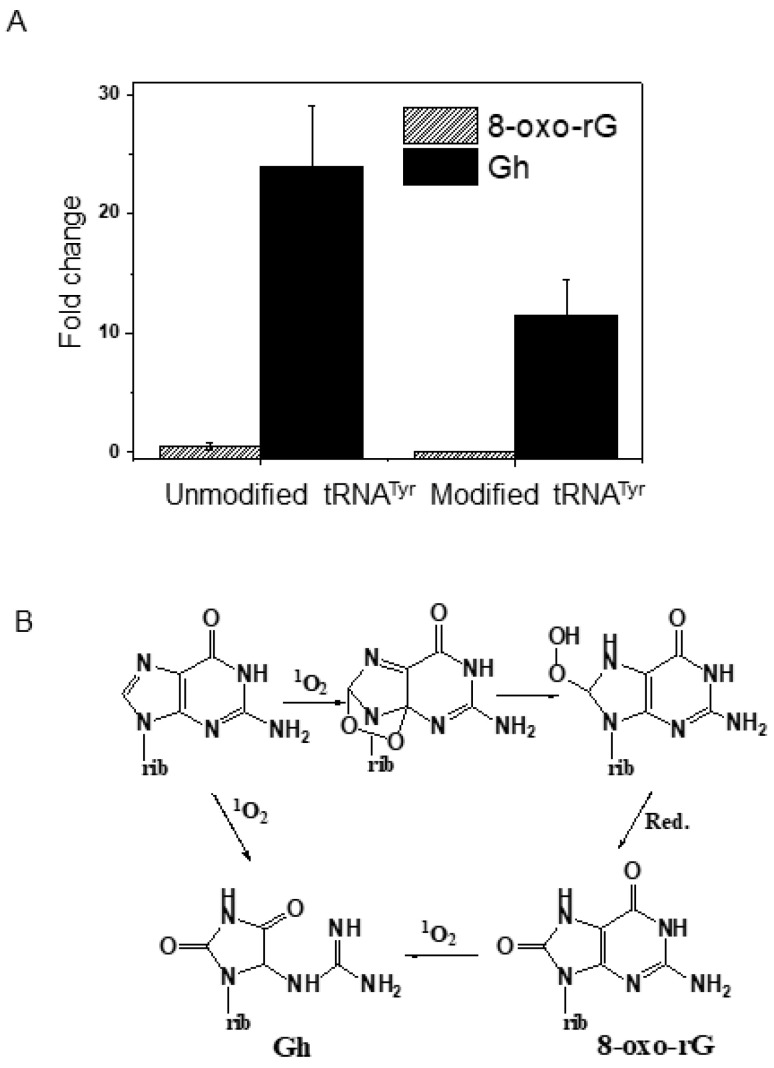
Guanosine oxidation levels in UV-exposed tRNA compared to unexposed samples. (**A**) An unmodified (in vitro transcript) and modified tRNA^Tyr^ were exposed to UVA in the presence of photosensitizer and fold change was computed by taking the ratio of the peak areas of guanosine oxidation products in UVA-exposed samples against that of unexposed samples. The signal for each oxidation product was normalized against the canonical guanosine in each sample. (**B**) Schematic flow of guanosine oxidation into 8-oxo-rG and guanidinohydantoin (Gh).

**Figure 4 biomolecules-10-01527-f004:**
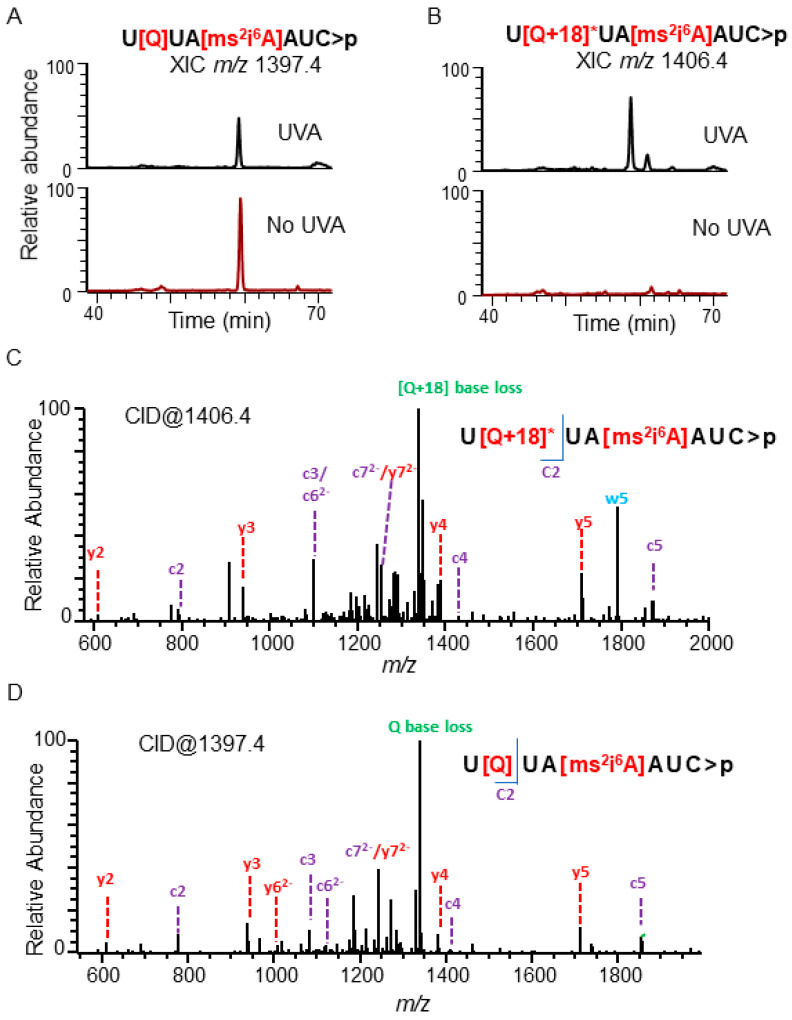
Chemical changes in the anticodon region of tRNA^Tyr^ from UVA-exposed and unexposed *E. coli* cells. (**A**) Comparative levels of modified oligonucleotide, U[Q]UA [ms^2^i^6^A]AUC>p (*m/z* 1397.4, position 33–40). (**B**) Comparative levels of modified oligonucleotide, U[Q+18]*UA [ms^2^i^6^A]AUC>p (*m/z* 1406.4), (**C**) MS/MS spectrum of *m/z* 1406.4 depicting the sequence informative product ion (color coded c and y fragment ions as described in Figure 2) pattern of oligomer, U[Q+18]*UA [ms^2^i^6^A]AUC>p (**D**) MS/MS spectrum of *m/z* 1397.4 depicting the sequence informative product ion pattern of oligomer, U[Q]UA [ms^2^i^6^A]AUC>p. Most abundant fragment ion correspond to the loss of Queuosine-Q or Q+18 and their positions are depicted in red font in the oligonucleotide.

**Figure 5 biomolecules-10-01527-f005:**
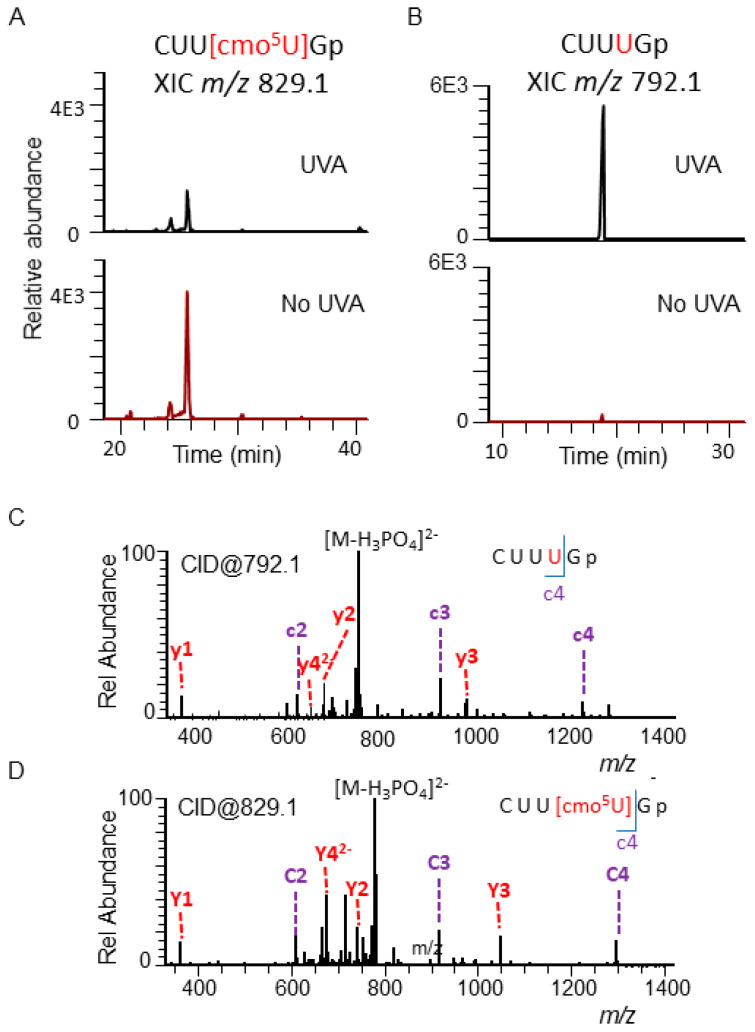
Chemical changes in the anticodon region of tRNA^Ala^ from UVA-exposed and unexposed *E. coli* cells. (**A**) Comparative levels of modified oligonucleotide, CUU[cmo^5^U]G (*m/z* 829.1, position 31–35). (**B**) Comparative levels of CUUUG (*m/z* 792.1, position 31–35). (**C**) MS/MS spectrum of *m/z* 792.1 depicting the sequence informative product ion (color coded c and y fragment) pattern of oligomer as described in Figure 2. (**D**) MS/MS spectrum of *m/z* 829.1 depicting the sequence informative product ion pattern of oligomer. The position of modification is shown in red font.

**Table 1 biomolecules-10-01527-t001:** UVA-induced alterations to tRNA^Tyr^ oligonucleotides.

tRNA Region	Unmodified RNA	Modified	
		In vitro Exposure	In vivo Exposure
Acceptor and D-arm junction	__	**UU**CCCGp **FapyG[U]*[U]***CCCGp	pGGUGGGG**UU**CCC>p
Anticodon	U[**Gh**]UAAAUC>p	U[**Q-99**]ψA[**ms^2^A**]Aψ C>p	U[**Q+18**]ψA[ms^2^i^6^A]AψC>p
Variable loop	UGCC[**Gh**]UCAUC>p AUC[**Gh**]AC>p UUCGAAG[**Gh**]UUC>p	UGCC[**Gh**]UCAUCGAC>p	__

Unmodified:.GGUGGGGUUCCCGAGCGGCCAAAGGGAGCAGACU**G**UAAAAUCUGCC**G**UCAUC**G**ACUUCGAAG**G**UUCGAAUCCUUCCCCCACCACCA. **Modified:**GGUGGG**G[s^4^U][s^4^U]**CCCGAGC[Gm]GCCAAAGGGAGCAGACU**[Q]**ψA**[ms^2^i^6^A]**AAψCUGCC**G**UCAUCGACUUCGAAGG[m^5^U]ψCGAAUCCUUCCCCCACCACCA. Locations of photooxidations are shown in bold faced font. Observed oligonucleotides are underlined in the sequence.

**Table 2 biomolecules-10-01527-t002:** UVA-induced effects on representative tRNA following cellular exposure.

tRNA	Modified Oligonucleotide	Potential UVA Alteration
Cys	U[s^4^U]AACAAAGp	UUAACAAAGp
Asn	ACU[Q]UU[t^6^A]AψCCGp	ACU[Q+18]UU[t^6^A]AψCCGp
Ala	CUU[cmo^5^U]Gp	CUUUGp
Glu	CCCU[mnm^5^s^2^U]UC[m2A]CGp	CCCUUUCACGp
Arg (ICG)	[m^2^A]ACCGp	AACCGp
Gln(UUG)	[m^2^A]ψACCGp	AψACCGp
Gln(CUG)	[m^2^A]ψψCCGp	-
His(GUG)	[m^2^A]ψψCCAGp	AψψCCAGp

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
