# Peer review of "Characterization of UVA-Induced Alterations to Transfer RNA Sequences"

_biomolecules, 2020, doi:10.3390/biom10111527_

Round 1

Reviewer 1 Report

In this manuscript, Addepalli and co-workers investigate the specific regions that are more vulnerable to photooxidation in tRNA upon irradiation at 370 nm (UVR) using the LC-MS/MS technique. It was observed that photooxidative damage occurs primarily to the anticodon and variable loop regions in both unmodified and modified sequences of tRNA. The type of oxidation product formed in the anticodon stem-loop region varies with modification, status, and whether the tRNA is exposed to UVR outside or inside cells.

This manuscript is of potential interest to the readers of Biomolecules and should be considered for publication after the authors take into full consideration the concerns enumerated below. Further review is necessary.

  1. The absorption spectra of the tRNA samples used for the irradiation experiments at 370 nm should be reported as Supplementary Material under the experimental conditions used for irradiation.
  2. Data in which the tRNA samples are irradiated at 370 nm without the presence of riboflavin (RF), but under equal irradiation intervals as those used in the presence of RF (20 min), should be reported as control experiments. In particular, some of the modifications such as the thiouracil derivatives absorb strongly at 370 nm and are known to photocrosslink with adenine and to form reactive oxygen species such as singlet oxygen in high yield under UVA excitation (see, for instance, https://doi.org/10.1039/C5CP04822B and https://doi.org/10.1021/ja961329g). Therefore, oxidatively-generated tRNA damage could result (in part) from direct UVA excitation in addition to RF photosensitization.
  3. Similar control, irradiation experiments as in #2 above (i.e., in the absence of RF) should be performed with cells (in vivo). All the sulfur-substituted nucleobases present in tRNA are known to form near-unity triplet yields in aqueous solution upon ultraviolet irradiation excitation and high yields of singlet oxygen that could result in oxidatively-generated damage to tRNA (https://doi.org/10.1111/php.12975).
  4. A wide-range of modifications are reported in the manuscript without defining their chemical structure (Ex. Cmo5U; mnm5s2U, t6A, etc.). All abbreviations must be explained the first instance that they are used.
  5. The font size of Figs. 1 and 2 is too small and should be increased. What the functional group (SOn-) in Fig. 2B stands for?
  6. Why NMR experiments before and after irradiation of the tRNA samples were not performed? These experiments can provide key structural information to better understand the role of structure on the selective formation of photooxidation products in the anticodon, variable loop, and thiolated regions.
  7. Throughout the discussion section, the authors should make an effort to limit speculative arguments (too many to enumerate all of them herein) that are completely supported by the experimental results obtained in this work. Such speculations detract from the main results presented in this work.
  8. Have authors verified that the digestion conditions used to hydrolyze the tRNA samples are not too harsh as to bias the detection (or lack of) of specific photooxidation products?
  9. There are many typographical errors in the reference section, which should be corrected.

Reviewer 2 Report

The manuscript entitled "Characterization of UVA-induced alterations to transfer RNA sequences" by Sun et al. reports the analysis of modified ribonucleosides in tRNA upon UVA exposure.

The authors identified (in vitro and in vivo) the different modified bases found in tRNA upon UVA irradiation using LC-MS/MS. The authors found new types of photoproducts and hot spots of modifications at particular positions.

The manuscript is well written and allows a perfect understanding of the experimental approach as well as the results.

The discussion clearly highlights the findings

This study represents an important contribution in the field of tRNA modifications

I have no major criticisms.

Reviewer 3 Report

  Liquid chromatography tandem mass spectrometry (LC-MS/MS)-based RNA  modification mapping approaches was applied to identify the transfer RNA (tRNA)  regions moat vulnerable to photooxidation. The nucleoside modifications were related to the oxidative damage induced by UV radiation. The type of oxidation product formed in the anticodon stem-loop region varied with the modification type, status, and whether inside or outside the cell during exposure.

Some points must clarify in the manuscript.

  1. UV radiation may cause DNA or RNA damage directly. Why 100 μM riboflavin was added when sample was exposed to UVA?
  2. The UVA exposure dose is 0.752 mJ/cm2 min for 20 min at room temperature. Different type of condition including strength and time of UVA exposure may lead to different results. Authors have to explain why the condition was used in this study.
  3. This study detected the sequence in t-RNA or E. coli. The results of the modification in tRNA or sequence verify in different condition or model. The skin cell model must apply in this study for further understand the modification of UV on tRNA.

Round 2

Reviewer 1 Report

The authors have made a genuine effort to answer all the concerns I had with the original version of this manuscript. The revised version is significantly improved and I now think that this manuscript should be considered for publication in the Biomolecules journal. There are still minor typographical errors, but I trust they will be handled during the proof-reding stage of the process. Further review is not necessary.

Author Response

I have made the required corrections.